# Consumers’ Preferences and Attitudes towards Plant-Based Milk

**DOI:** 10.3390/foods13010002

**Published:** 2023-12-19

**Authors:** Wenfan Su, Yu Yvette Zhang, Songhan Li, Jiping Sheng

**Affiliations:** 1School of Agricultural Economics and Rural Development, Renmin University of China, Beijing 100872, China; suwenfan15@126.com (W.S.); lisonghan@sinochem.com (S.L.); 2Department of Agricultural Economics, Institute for Advancing Health through Agriculture (IHA) Affiliate Member, Texas A&M University, College Station, TX 77843, USA; yzhang@tamu.edu

**Keywords:** plant-based milk, willingness to pay, consumer preference, environmental awareness, health consciousness, food neophobia

## Abstract

Plant-based milk (PBM) has become increasingly popular due to its environmental sustainability, health benefits, ingredient abundance, and unique taste. This study aims to identify the main factors that affect consumer preferences and attitudes towards PBM, and to examine the effect of consumer attitudes including environmental awareness, health consciousness, and food neophobia on WTP. We use the double-bounded dichotomy choice (DBDC) method to calculate consumers’ willingness to pay (WTP) for PBM. We find that the appearance, taste, nutritional value, and environmental benefits of PBM significantly increase consumers’ WTP for it. Consumers with high environmental awareness are more likely to perceive PBM as environmentally friendly and are willing to pay a higher price for it. Consumers with high health consciousness tend to value the environmental benefits of PBM and prioritize purchase convenience, as it aligns with their health-conscious lifestyle, leading to a higher WTP for PBM. The results of our study can help design effective strategies to market plant-based milk and develop sustainable and healthy food systems.

## 1. Introduction

The need to build a low-carbon sustainable food system is becoming increasingly urgent in the face of climate change, environmental pollution, and the resulting crises. One way to reduce greenhouse gas emissions from diets and alleviate pollution caused by animal husbandry is to substitute animal-based food with plant-based food [1,2]. Plant-based milk (PBM), such as oat, almond, rice, or soy milk, has become a popular alternative to dairy milk due to its environmental sustainability and health benefits [3]. PBM production emits only one third of the greenhouse gases compared to dairy milk production and requires far less land [4,5,6]. Unlike dairy milk, PBM does not contain lactose or dairy milk proteins. It is a better choice for those with dairy milk allergies or lactose intolerance [7], particularly in African American and Asian ethnicities, where lactose intolerance rates range from 75% to 95% [8,9] Additionally, it is cholesterol-free, rich in vitamins and minerals, and beneficial for lactose intolerance, and can help reduce cholesterol, diabetes or other diseases [10,11,12]. Hence, PBM is a healthier functional beverage when animal milk needs to be replaced in the diet.

The consumption of PBM has seen a significant increase in the global food industry in recent years due to heightened awareness among consumers about health and environmental issues [13]. The global market share of PBM has risen from below 10% in 2015 to over 13% by 2018 [14]. Remarkably, the Chinese plant-based protein beverage market witnessed a staggering growth rate of 800% in 2020 [15]. With its deep-rooted tradition of plant-based food consumption, China serves as one of the largest platforms for milk alternatives [16]. However, despite this positive trend, the PBM market faces significant challenges. Notably, the PBM sector grapples with high product price points relative to traditional dairy products, which could be a deterrent for wider consumer adoption. Furthermore, consumer acceptance of new food products, such as PBM, is often hindered by food neophobia, a phenomenon that could significantly influence the success of PBM in the market. Lastly, while the health and environmental benefits of PBM are extensively promoted, it remains ambiguous how greatly these factors impact consumer purchase decisions and their WTP. To address these complexities and contribute to the broader understanding of the PBM market dynamics, we embarked on this research journey. The first dimension of our research focuses on eliciting the willingness to pay (WTP) of consumers in China for PBM via the double-bounded dichotomous choice contingent valuation method (DBDC-CVM). The second dimension is to examine the effect of consumer attitudes including environmental awareness, health consciousness and food neophobia on WTP. This study intends to unveil the intricate interplay between these factors, providing valuable insights for stakeholders in the PBM market to navigate, strategize, and flourish amidst these challenges.

## 2. Literature Review

### 2.1. Plant-Based Milk (PBM) Market

The PBM market has undergone substantial growth, becoming a significant sector within the global food industry, expanding its product range beyond traditional soy and almond milk to diverse sources such as oats, hemp, and rice [17,18]. The rise in PBM popularity reportedly corresponds with a decrease in dairy milk sales [19]. Key drivers of this growth of PBM include the upgrading of taste, nutrient fortification, and increasing environmental and health awareness among consumers [20,21,22]. Public values related to environmental impact and animal welfare are notable influencers of demand for PBM [23,24]. The development of the plant-based milk market is additionally influenced by consumer perception, labeling claims, regulatory frameworks, and novel processing technologies, alongside sociodemographic factors [25,26,27]. Despite the robust growth of the PBM market, driven by health consciousness, environmental awareness, dietary restrictions, cultural acceptance, and industry innovation, the dynamic nature of the PBM market necessitates ongoing research to comprehend evolving trends.

### 2.2. Consumer Attitudes towards Plant-Based Milk (PBM)

Consumers’ environmental awareness, health consciousness, and food neophobia reflect their perceptions or attitudes related to environmental practices, health behaviors, and food choices. These attitudes towards PBM have been central to understanding the growth and acceptance of PBM. Several studies have examined these aspects individually and collectively to build a comprehensive understanding of consumer behavior. The connection between consumer environmental awareness and the consumption of PBM has been well-established in the literature. Vermeir and Verbeke [28] found that consumers with higher concerns for the environment were more likely to consume plant-based food products. More recently, Hartmann and Siegrist [29] expanded on this by demonstrating that consumers’ environmental consciousness directly influences their WTP for PBM, thus reinforcing the role of environmental awareness in driving the PBM market. McCarthy et al. [30] discovered that exclusive nondairy plant-based beverage consumers were driven by a desire to reduce animal product intake, concerns about animal welfare, and the perception of a lower environmental impact compared to dairy milk.

Health consciousness, specifically the perception of health benefits, plays a significant role in shaping consumer attitudes towards PBM. Despite the challenges associated with production and sensory profiles, PBM is lauded for its potential health benefits, largely attributed to its lower energy production requirements and the possibility of nutritional customization [31]. Jallinoja et al. [32] demonstrate a prevalent consumer belief that PBM is nutritionally superior to dairy milk, thereby enhancing the appeal of these alternatives. Hoek et al. [33] also demonstrate that health perception also greatly influences the readiness to replace meat with plant-based meat. Prytulska et al. [34] corroborate the significant impact of health consciousness on favoring PBM. Moreover, Martínez-Padilla et al. [35] found that positive perceptions of PBM’s naturalness, health benefits, and nutritional parity to cow’s milk bolstered its consumption, whereas perceptions of PBM as excessively processed or synthetic deterred consumption. Collectively, these studies underscore the pivotal role of health consciousness in determining PBM consumption trends.

Moreover, food neophobia, or the reluctance to try new or unfamiliar foods [36], is another crucial factor in consumer acceptance of PBM. Ritchey et al. [37] examined the impact of food neophobia on the acceptance of PBM, concluding that higher levels of neophobia were associated with a lower likelihood of PBM consumption. Faber and Petersen [35] found that consumers who perceive PBM as natural, healthy, tasty, or nutritionally equivalent to dairy milk are more likely to consume it, whereas those who view it as highly processed or artificial are less likely to do so.

The literature has provided a multi-faceted understanding of consumer attitudes towards PBM, with environmental awareness, health consciousness, and food neophobia emerging as key influencers. However, the interplay between these factors and their collective impact on PBM consumption warrants further exploration to inform market strategies and interventions effectively.

### 2.3. Consumers’ Willingness to Pay for Plant-Based Milk (PBM)

The concept of WTP is a central principle in understanding consumer behavior, particularly in the context of environmentally sustainable and health-focused products such as PBM. Various studies have explored this concept, with methodologies such as the contingent valuation method and choice experiment being widely used. The contingent valuation method (CVM) differs from choice experiments in that it does not require respondents to navigate a series of trade-offs between diverse product attributes and prices. Instead, CVM poses a simplified, singular question about WTP, thereby potentially reducing the cognitive burden on respondents and enhancing the quality of the collected data [38]. Compared with the single-bounded method, DBDC method can provide more information about the “real” WTP of participants and yield more accurate WTP [39]. Applying the DBDC-CVM, Gracia et al. [40] investigated consumer WTP for organic milk, finding that consumers were willing to pay a premium for organic products due to perceived health benefits. Their work set a precedent for similar applications of DBDC-CVM in the PBM context. More recently, Li et al. [41] expanded on this by investigating how the origin of PBM influences WTP. Using the DBDC-CVM, they found that consumers were willing to pay more for locally sourced PBM, highlighting the role of product origin in shaping consumer WTP.

While these models have proven valuable, they are not without potential issues, including point bias and incentive incompatibility. Several studies have endeavored to address these challenges empirically. This bias arises when respondents focus on the initial bid value, potentially skewing results. Iterative bidding has been proposed as a method to mitigate point bias, which has since been adopted in numerous studies. Incentive incompatibility, another common issue, refers to the potential for strategic behavior by respondents, who may misrepresent their true WTP in an effort to influence the final price of a product. Carson and Groves [42] addressed this by emphasizing the importance of making clear to respondents that their answers could potentially influence real outcomes, thus encouraging truthful responses. To further enhance the robustness of DBDC-CVM model results, some studies have combined the model with other methods, such as integrated choice and latent variable models [43].

## 3. Survey

Our study was conducted in 2021 in the Jing-Jin-Ji metropolitan region of China, which is the largest urbanized megalopolis in northern China. This region encompasses the municipalities of Beijing and Tianjin, as well as the Hebei province [44]. We sent out 900 questionnaires using the online survey platform Wenjuanxing (https://www.wjx.cn, accessed on 28 January 2021), with 300 questionnaires in each region. We obtained 819 valid responses. To ensure the quality of the survey, we set a minimum response time of 3 s per question to prevent participants from rushing through the questions. Additionally, we implemented technical identification measures, including IP and device tracking, to prevent duplicate responses from the same user and avoid repeated sampling. Participants who do not consume milk or milk alternatives were also included in the survey. Such participants, representing a potential market for PBM, may offer unique insights into the barriers to the consumption of PBM products.

The study sample consisted of 59% female and 41% male participants. The average age of participants was 32.4 years. A majority of the participants hold a college degree, indicating a higher level of education compared to the broader Chinese population. The income levels of participants are fairly distributed across three categories. Dietary preferences show that 7% of participants adhere to a vegan or vegetarian diet and 9% report lactose intolerance. In this study, we conducted a survey of consumer preferences for six attributes of PBM using the Likert scale, including appearance, taste, nutritional value, environmental benefits, price, and purchase convenience. Basic information about PBM, such as its nutritional health and environmental benefits, is provided. Table 1 shows consumer characteristics and their preferences for product attributes. The results show that the nutritional value of PBM was the most preferred by consumers, followed by product price and purchase convenience, while the appearance and taste were less preferred.

We also measured consumers’ attitudes, including environmental awareness, health consciousness, and food neophobia (Table 2). The scale for measuring environmental awareness was adapted from Lindeman and Vaananen [45], Kollmuss and Agyeman [46], and Wesley et al. [47]. We selected key elements of environmental awareness such as animal welfare, eco-friendly food production, packaging, and the maintenance of an ecological balance. The scale for health consciousness was adapted from Gabriel et al. [48], and includes items related to personal health awareness and perceived control of health. The food neophobia scale was adapted from Demattè et al. [49] and Flight et al. [50]. All statements were rated by each individual on a 7-point scale from “strongly disagree = 1” to “strongly agree = 7”. Summary statistics are shown in Table 2. Our scales had good reliability and validity (demonstrated in Appendix A). The individual scores for environmental awareness, health consciousness, and food neophobia were computed as the average of ratings given to each statement. Based on the scores, participants were divided into three groups according to the level of environmental awareness (low, medium, and high), health consciousness (low, medium, and high), and food neophobia (low, medium, and high). This categorization was undertaken by dividing the distribution of scores into three equal quantiles, effectively sorting the data into these three groups.

Furthermore, we utilized the DBDC-CVM method to measure consumers’ WTP for PBM. Each participant engaged in two rounds of bidding: participants responded to a first amount and then faced a second question involving another amount, higher or lower depending on the response to the first question [39]. Our survey design was informed by preliminary market research, which indicated that the average price for an assortment of PBM products across a variety of types (including peanut milk, walnut milk, and coconut milk) and brands in China’s 2020 market was approximately CNY 7.45 per 250 mL. In addition, we conducted focus group discussions with 12 participants, chosen to represent a broad and diverse spectrum of consumer demographics including age, gender, socioeconomic background, and prior exposure to PBM products. During these discussions, participants were presented with a range of PBM product prices derived from our market research. They were encouraged to discuss their thoughts on these prices, their personal WTP within this range, and potential influencing factors. We introduced our DBDC scheme and discussed potential starting prices. Participants expressed general acceptance of our proposed starting price of CNY 7, marginally below the average market price, considering it a realistic purchasing price for PBM. They suggested that significantly lower starting prices (such as CNY 5 or 6) could skew the survey responses by not accurately representing the current market situation. These focus group discussions were instrumental in selecting CNY 7 as our survey’s starting price, enhancing the validity of our research design by realistically reflecting consumer price acceptance and rejection for PBM products. Each participant was first randomly given a price (CNY 7, 8, 9, or 10) and asked if they were willing to accept this starting (first) price. If they accepted it, they would further receive a higher price; if they rejected it, they would subsequently receive a lower price (Table 3). Summary statistics of consumer WTP in the DBDC survey are presented in Table 4.

## 4. Analytical Framework

Based on the stochastic expected utility theory [51], we constructed the following utility of choosing PBM or dairy milk:(1)$$U_{1}=\alpha_{1}+{\beta^{\prime }}_{1}X+\lambda_{1}BID+\varepsilon_{1}$$
(2)$$U_{0}=\alpha_{0}+{\beta^{\prime }}_{0}X+\varepsilon_{0}$$
assuming that the utility of consumers choosing PBM can be expressed as *U*_1_, and the utility of choosing dairy milk can be expressed as *U*_0_. *X* represents the key explanatory factors that affect consumer utility, such as consumers’ preferences for product attributes. *BID* is used to indicate how much more consumers are willing to pay for PBM compared to dairy milk, i.e., paying a premium.

When $U_{1}$ > ${\;U}_{0}$, consumers will choose PBM, while when $U_{1}$ < $U_{0}$, consumers will not choose PBM. The difference between the two utilities is calculated as follows:(3)$$\begin{array}{c} \Delta U=U_{1}-U_{0}=\left(\alpha_{1}-\alpha_{0}\right)+{\left(\beta_{1}-\beta_{0}\right)}^{\prime }X+\lambda_{1}BID+(\varepsilon_{1}-\varepsilon_{0})=\Delta \alpha +\\ {\left(\Delta \beta \right)}^{\prime }X+\lambda_{1}BID+\Delta \varepsilon \end{array}$$
where *ε* follows the Weibull distribution. The differences in the random variables are the logistic distribution [52,53]. The probability of choosing PBM is
(4)$$\begin{array}{c} P\left(Y=1\right)=P\left(\Delta U>0\right)=P[\Delta \varepsilon >-\Delta \alpha +{\left(\Delta \beta \right)}^{\prime }X+\lambda_{1}BID)]=\\ {\left[1+\exp\left(-\Delta U\right)\right]}^{-1}. \end{array}$$

The maximum likelihood estimation model is
(5)$$\ln\left[\frac{P\left(Y=1\right)}{1-P\left(Y=1\right)}\right]=\Delta \alpha +{\left(\Delta \beta \right)}^{\prime }X+\lambda_{1}BID.$$

Considering that the contingent valuation method (CVM) performs well in cases where the hypothetical situation is similar to a familiar market choice situation [54], we used DBDC method to measure consumers’ WTP for PBM.
(6)$$\alpha_{1}+{\beta^{\prime }}_{1}X+\lambda_{1}BID+\varepsilon_{1}=\alpha_{0}+{\beta \prime }_{0}X+\varepsilon_{0}$$

When
(7)$$E\left(\varepsilon_{1}\right)=E(\varepsilon_{0})\;,$$

Then
(8)$$E\left(BID\right)=E\left(WTP\right)=\frac{\Delta \alpha +({\Delta \beta )}^{\prime }E(X)}{\lambda_{1}}$$

Then, we defined $y_{i}^{1}$ and $y_{i}^{2}$ as binary variables in response to two closed questions, and then the probability of an individual answering “yes” to the first question and “no” to the second question could be expressed as
(9)$$\mathrm{Pr}\left(y_{i}^{1}=1,\;y_{i}^{2}=0|z_{i}\right)=Pr(y,n)\;.$$
where $z_{i}$ represents the respondent. $y_{i}^{1}$ and $y_{i}^{2}$ represent the answers to the two questions.

The *WTP* can be calculated as
(10)$${WTP}_{i}\left(z_{i},u_{i}\right)=z_{i}^{\prime }\beta +u_{i},u_{i}\sim N(0,\sigma^{2}).$$

Let *m*_1_ and *m*_2_ be the set prices in the two questions, respectively. According to the answer of “yes” or “no” to the two questions, there are four cases, respectively, and their probabilities are shown as follows:

The first case: when $y_{i}^{1}=1,\;y_{i}^{2}=$0, then
(11)$$\begin{array}{c} \mathrm{Pr}\left(y,n\right)=\mathrm{Pr}\left(m_{1}\leq WTP\leq m_{2}\right)=\mathrm{Pr}\left(m_{1}\leq z_{i}^{\prime }\beta +u_{i}\leq m_{2}\right)=\mathrm{Pr}\left(\frac{m_{1}-z_{i}^{\prime }\beta }{\sigma }\leq \frac{u_{i}}{\sigma }\leq \frac{m_{2}-z_{i}^{\prime }\beta }{\sigma }\right)\\ =\Phi \left(\frac{m_{2}-z_{i}^{\prime }\beta }{\sigma }\right)-\Phi \left(\frac{m_{1}-z_{i}^{\prime }\beta }{\sigma }\right)=\Phi \left(z_{i}^{\prime }\frac{\beta }{\sigma }-\frac{m_{1}}{\sigma }\right)-\Phi \left(z_{i}^{\prime }\frac{\beta }{\sigma }-\frac{m_{2}}{\sigma }\right) \end{array}$$

The second case: when $y_{i}^{1}=1,\;y_{i}^{2}=1$,
(12)$$\begin{array}{c} \mathrm{Pr}\left(y,y\right)=\mathrm{Pr}\left(WTP\geq m_{1},WTP\geq m_{2}\right)=\mathrm{Pr}\left(z_{i}^{\prime }\beta +u_{i}\geq m_{1},{z_{i}^{\prime }\beta +u_{i}\geq m}_{2}\right)\\ =\mathrm{Pr}\left(z_{i}^{\prime }\beta +u_{i}\geq m_{1}|z_{i}^{\prime }\beta +u_{i}\geq m_{2}\right)*\mathrm{Pr}\left(z_{i}^{\prime }\beta +u_{i}\geq m_{2}\right) \end{array}$$

Because $m_{2}>m_{1},$ and
(13)$$\mathrm{Pr}\left(z_{i}^{\prime }\beta +u_{i}\geq m_{1}|z_{i}^{\prime }\beta +u_{i}\geq m_{2}\right)=1$$

Then,
(14)$$\mathrm{Pr}\left(y,y\right)=\mathrm{Pr}\left(u_{i}\geq m_{2}-z_{i}^{\prime }\beta \right)=1-\Phi \left(\frac{m_{2}-z_{i}^{\prime }\beta }{\sigma }\right)=\Phi \left(z_{i}^{\prime }\frac{\beta }{\sigma }-\frac{m_{2}}{\sigma }\right)$$

The third scenario: when $y_{i}^{1}=0,\;y_{i}^{2}=1$, then
(15)$$\begin{array}{c} \mathrm{Pr}\left(y,n\right)=\mathrm{Pr}\left(m_{2}\leq WTP\leq m_{1}\right)=\mathrm{Pr}\left(m_{2}\leq z_{i}^{\prime }\beta +u_{i}\leq m_{1}\right)=\mathrm{Pr}\left(\frac{m_{2}-z_{i}^{\prime }\beta }{\sigma }\leq \frac{u_{i}}{\sigma }\leq \frac{m_{1}-z_{i}^{\prime }\beta }{\sigma }\right)\\ =\Phi \left(\frac{m_{1}-z_{i}^{\prime }\beta }{\sigma }\right)-\Phi \left(\frac{m_{2}-z_{i}^{\prime }\beta }{\sigma }\right)=\Phi \left(z_{i}^{\prime }\frac{\beta }{\sigma }-\frac{m_{2}}{\sigma }\right)-\Phi \left(z_{i}^{\prime }\frac{\beta }{\sigma }-\frac{m_{1}}{\sigma }\right). \end{array}$$

The fourth scenario: when $y_{i}^{1}=0,\;y_{i}^{2}=0$, then
(16)$$\begin{array}{l} \mathrm{Pr}\left(n,n\right)=\mathrm{Pr}\left(WTP<m_{1},WTP<m_{2}\right)=\mathrm{Pr}\left(z_{i}^{\prime }\beta +u_{i}<m_{1},{z_{i}^{\prime }\beta +u_{i}<m}_{2}\right)=\mathrm{Pr}\left({z_{i}^{\prime }\beta +u_{i}<m}_{2}\right)\\ \;\;\;\;\;=\Phi \left(\frac{m_{2}-z_{i}^{\prime }\beta }{\sigma }\right)=1-\Phi \left(z_{i}^{\prime }\frac{\beta }{\sigma }-\frac{m_{2}}{\sigma }\right) \end{array}$$

The maximum likelihood estimation function is:(17)$$\begin{array}{c} \sum_{i=1}^{N}[d_{i}^{yn}\ln\left(\Phi \left(z_{i}^{\prime }\frac{\beta }{\sigma }-\frac{m_{1}}{\sigma }\right)-\Phi \left(z_{i}^{\prime }\frac{\beta }{\sigma }-\frac{m_{2}}{\sigma }\right)\right)+d_{i}^{yy}\ln\left(\Phi \left(z_{i}^{\prime }\frac{\beta }{\sigma }-\frac{m_{2}}{\sigma }\right)\right)+d_{i}^{ny}\ln(\Phi \left(z_{i}^{\prime }\frac{\beta }{\sigma }-\frac{m_{2}}{\sigma }\right)-\\ \Phi \left(z_{i}^{\prime }\frac{\beta }{\sigma }-\frac{m_{1}}{\sigma }\right))+d_{i}^{nn}\ln\left(1-\Phi \left(z_{i}^{\prime }\frac{\beta }{\sigma }-\frac{m_{2}}{\sigma }\right)\right)], \end{array}$$
where $d_{i}^{yn},\;d_{i}^{yy},d_{i}^{ny},d_{i}^{nn}$ are variables with a value of 1 or 0 according to the relevant situation of each individual; that is, a given individual contributes to the logarithm of the likelihood function in one of four parts.

The resulting parameter $\hat{\beta }$ is used to estimate the WTP:(18)$$E(\mathrm{WTP})=z_{i}^{\prime }\hat{\beta }.$$

These are the expected WTP values that reflect variations in *z_i_* and the actual individual *WTP* may vary more substantially around these estimates. After calculating the WTP for the *PBM*, we further identified key influencing factors which would play important roles in shaping consumers’ preferences and *WTP* for *PBM*. These factors include consumer preferences for appearance, taste, nutritional value, environmental benefits, price, and convenience of the *PBM*. We used a multiple linear regression model for econometric analysis.
(19)$$Y_{i}=\beta_{0}+a_{ij}X_{ij}+b_{ik}Z_{ik}+c_{ig}H_{ig}+\mu$$
where $Y_{i}$ is participant *i*′ WTP for PBM calculated via DBDC-CVM. $\beta_{0}$ is the intercept term.$\;X_{ij}$ is participant *i*′ preferences for product attributes *j* measured by a 7-point Likert scale, and product attributes *j* include appearance, taste, nutritional value, environmental benefits, price, and convenience of the PBM. $Z_{ik}$ is participant *i*′ attitude *k*, including environmental awareness, health consciousness and food neophobia; $H_{ig}$ is participant *i*′ with demographic characteristic *g* as the control variables; ${a_{ij}b}_{ik},\;c_{ig}$ are coefficients of each variable. *μ* is the residual.

## 5. Results

### 5.1. Willingness to Pay (WTP) for Plant-Based Milk (PBM) and Its Influencing Factors

In our sample, the average WTP for PBM was CNY 7.919 per 250 mL, with a standard deviation of 0.979. The WTP values, ranging from CNY 4.827 to 11.321, followed a normal distribution. We utilized ordinary least squares (OLS) regression in Model 1 to explore the effect of consumer preferences and characteristics on WTP for PBM. Model 2 further extended this analysis by incorporating consumer attitudes (Table 5). In model 1, the coefficients of appearance, sensory attributes, nutritional value, and environmental benefits are positive at the 1% significant level. It indicates that these factors are positively associated with WTP for PBM. Consumers’ preference for environmental benefits has the greatest positive effect on WTP. Conversely, the coefficients of purchase convenience and the importance of price are negative at 1% significant level, which means that these two factors have negative impacts on WTP. Consequently, consumers who prioritize the appearance, taste, nutritional value, and environmental benefits of PBM are more likely to pay a higher price for PBM, while those who are sensitive to price or prioritize convenience may be less inclined to pay a premium. In model 2, the results show that regarding consumer attitudes, environmental awareness and food neophobia have a significant positive effect on WTP.

As for consumer characteristics in model 1, being male is associated with a decrease in WTP by CNY 0.642. Each additional year of age is associated with a decrease in WTP by CNY 0.01. This might suggest that younger consumers are more willing to pay for PBM. Monthly income and education level have a positive impact on WTP for PBM. Living in an urban area increases WTP by CNY 0.3. Health-related factors, such as lactose intolerance, vegetarianism, or specific dietary requirements, also contributed to a higher WTP. Being lactose intolerant increases WTP by CNY 0.244. This makes sense as those with lactose intolerance are more likely to consume plant-based milk. Being a vegetarian increases WTP by CNY 1.086. Consumers who suffer from hyperglycemia, hypertension, or hyperlipidemia are willing to pay an additional CNY 0.154 for plant-based milk compared to those who do not suffer from these health conditions. This finding may indicate that consumers with these health conditions perceive plant-based milk as a healthier alternative to traditional dairy products or as part of dietary management for their conditions. Individuals in sustainable occupations related to medicine, health, and environmental protection are willing to pay an additional CNY 1.016 for plant-based milk compared to their counterparts in other occupations. This finding may suggest that those working in fields with a direct link to health and sustainability have a higher appreciation for plant-based milk, which is often marketed as a healthy and environmentally friendly product. The purchasing behavior of relatives and friends positively influenced consumers’ WTP for PBM, indicating the peer effect of recommendations from trusted sources.

### 5.2. The Effect of Consumer Attitudes on Willingness to Pay (WTP)

#### 5.2.1. Environmental Awareness

We conducted grouping regressions to examine the moderating effect of consumer attitudes on product attribute preference and WTP. Figure 1 depicts the distribution of WTP stratified by varying degrees of environmental awareness, represented in terms of price (CNY per 250 mL). The observed values below 4.827 could be due to different individual-specific random effects or error terms, which are not captured by the expected WTP (*z_i_β*). The results indicate a positive correlation between environmental awareness and WTP. A subsequent regression analysis investigating the influence of product attribute preferences among these environmental awareness groups found that heightened environmental awareness was associated with an increased emphasis on the environmental benefits of PBM (Table 6). Notably, the significance of price as a negative factor remained consistent across all levels of environmental awareness.

#### 5.2.2. Health Consciousness

Figure 2 presents the distribution of WTP across different levels of health consciousness, which shows a positive association between health consciousness and WTP. Additional regression analysis demonstrated that highly health conscious consumers tend to prioritize environmental benefits and purchase convenience over product appearance, thereby leading to an increased WTP for plant-based milk (Table 7).

#### 5.2.3. Food Neophobia

As shown in Figure 3, despite their general aversion to unfamiliar foods, individuals with high food neophobia are willing to pay more for PBM compared to individuals with low or medium levels of food neophobia. For consumers who exhibit a high level of food neophobia, those who prioritize taste and purchase convenience are likely to have a higher WTP for PBM (Table 8).

## 6. Discussion

Our research scrutinized the factors influencing Chinese consumers’ WTP for PBM, utilizing DBDC-CVM. The average WTP was determined as CNY 7.919 per 250 mL. Preferences for product attributes—appearance, sensory aspects, nutritional value, and environmental benefits—positively correlated with WTP, mirroring the findings of Rombach et al. [55], who identified environmental, food safety, health, and sustainability concerns as influential predictors of consumer behaviors. Among the attributes, environmental benefits exerted the most substantial positive impact on WTP, a finding that is consistent with Rombach et al.’s study [13]. Their research emphasized the significant effect of animal welfare considerations, a component of environmental consciousness, on the WTP premium for PBM. Conversely, purchase convenience and price sensitivity inversely correlated with WTP.

The determinants of WTP for PBM vary across demographic groups. Our research indicates that both female and younger consumers tend to exhibit a higher WTP for PBM. Additional factors, including elevated monthly income, advanced education level, and urban residency, also positively influence WTP. Dietary choices are intricately linked to ethical, health, and environmental concerns, as supported by the existing literature [56]. Correspondingly, our study identifies that individuals with lactose intolerance, vegetarians, and those with specific dietary needs express an elevated WTP for PBM. The lactose-free attribute of PBM allows lactose-intolerant individuals to sidestep the discomfort associated with dairy consumption, potentially leading to their higher WTP [57,58]. Vegetarians may demonstrate an increased WTP for PBM due to the alignment of these products with their dietary restrictions and ethical convictions [59]. Prior studies suggest that vegan consumers exhibit a significant preference for environmentally friendly products [60]. Additionally, individuals engaged in health and environmental protection sectors display an elevated WTP for PBM, implying a correlation between occupational principles, personal beliefs, and PBM valuation. It is noteworthy that trusted peer influence exerts a positive effect on consumers’ WTP for PBM, emphasizing the significance of word-of-mouth marketing for novel foods [61].

A positive association was identified between environmental awareness, health consciousness, and WTP for PBM. Consumers with a heightened sense of environmental awareness exhibited a higher WTP for PBM, largely attributable to their recognition of its environmental advantages. This behavior is consistent with the established notion that ethical motivations, such as environmental sustainability, often underpin the decision to purchase plant-based foods [62]. Health-conscious consumers, those who intentionally adopt dietary habits to enhance or maintain their well-being, also demonstrated higher WTP for PBM. These consumers typically adhere to a holistic health perspective, acknowledging the interconnectedness of environmental health and individual health outcomes [63,64]. They value PBM not only for its environmental benefits, but also for the convenience it offers in terms of purchasing and integration into daily dietary practices. The expanding PBM market has improved the accessibility and availability of these products, catering to health-conscious consumers who meticulously balance their health objectives with other life commitments. Interestingly, individuals with high food neophobia or resistance to novel foods are more willing to pay for the taste and convenience of PBM. The positive effect of preference for taste on WTP among high-neophobia individuals suggests that despite their aversion to new foods, they are willing to pay more if the taste is appealing. This aligns with the literature stating that taste is a key factor in food choice, even among neophobic consumers [65]. The positive effect of preference for convenience on WTP among high-neophobia individuals indicates that convenience significantly increases their WTP for PBM. This could be because convenience reduces the perceived risk and effort associated with trying new foods [66,67].

This study offers several managerial and policy implications for industry managers and marketers. First, the research shows the importance of consumers’ environmental consciousness and health awareness in their WTP for PBM. Producers can exploit these insights by designing marketing strategies that underscore the environmental sustainability and health benefits of their products, specifically targeting consumers who regard these attributes highly. Second, policy initiatives should aim to mitigate food neophobia through public education about novel foods. Strategies could include taste-tests, cooking demonstrations, and the distribution of easy-to-understand recipes, aiming to familiarize consumers with PBM and potentially boost its acceptance. Third, given the significant potential of PBM to contribute to a healthier and more sustainable food system, policymakers could contemplate providing support to the PBM industry. This could take the form of subsidies, R&D grants, or regulations favorable to the production and marketing of PBM.

This study has some limitations. One is the lack of a control group. Our results only imply consumers’ preferences for PBM, but not the comparisons between PBM and other products such as dairy milk. Another limitation of our study is that the CVM method utilized hypothetical purchasing scenarios, which might overestimate WTP estimates. Future studies should take into account these factors to better understand consumers’ preferences and WTP for the PBM products. Longitudinal studies could track changes in consumer attitudes over time. It would also be beneficial to compare stated preferences and willingness to pay with actual purchasing behavior to reinforce the findings. As alternative proteins evolve, exploring consumer responses to options like insect-based proteins or lab-grown meat could provide valuable insights into the future protein market. Additionally, understanding the influence of specific information about PBM’s health and environmental benefits on consumer perception could be instrumental in shaping educational and marketing strategies.

## 7. Conclusions

This paper examined consumers’ preferences and WTP for PBM based on the DBDC-CVM method. The findings imply that attributes such as the appearance, taste, nutritional value, and environmental benefits of PBM, alongside consumers’ environmental awareness, health consciousness, and food neophobia, significantly influence their preferences and WTP for PBM. Consumers with high environmental awareness are more likely to perceive PBM as environmentally friendly and are willing to pay a higher price for it. Similarly, consumers with high health consciousness tend to value the environmental benefits of PBM and prioritize purchase convenience, as it aligns with their health-conscious lifestyle, leading to a higher WTP for PBM. Interestingly, consumers with high food neophobia exhibit greater WTP for PBM than those with low or medium food neophobia, primarily due to their elevated taste expectations and the importance they place on purchasing convenience. These results underscore the potential role of PBM in constructing a healthy and sustainable food system.

## Figures and Tables

**Figure 1 foods-13-00002-f001:**
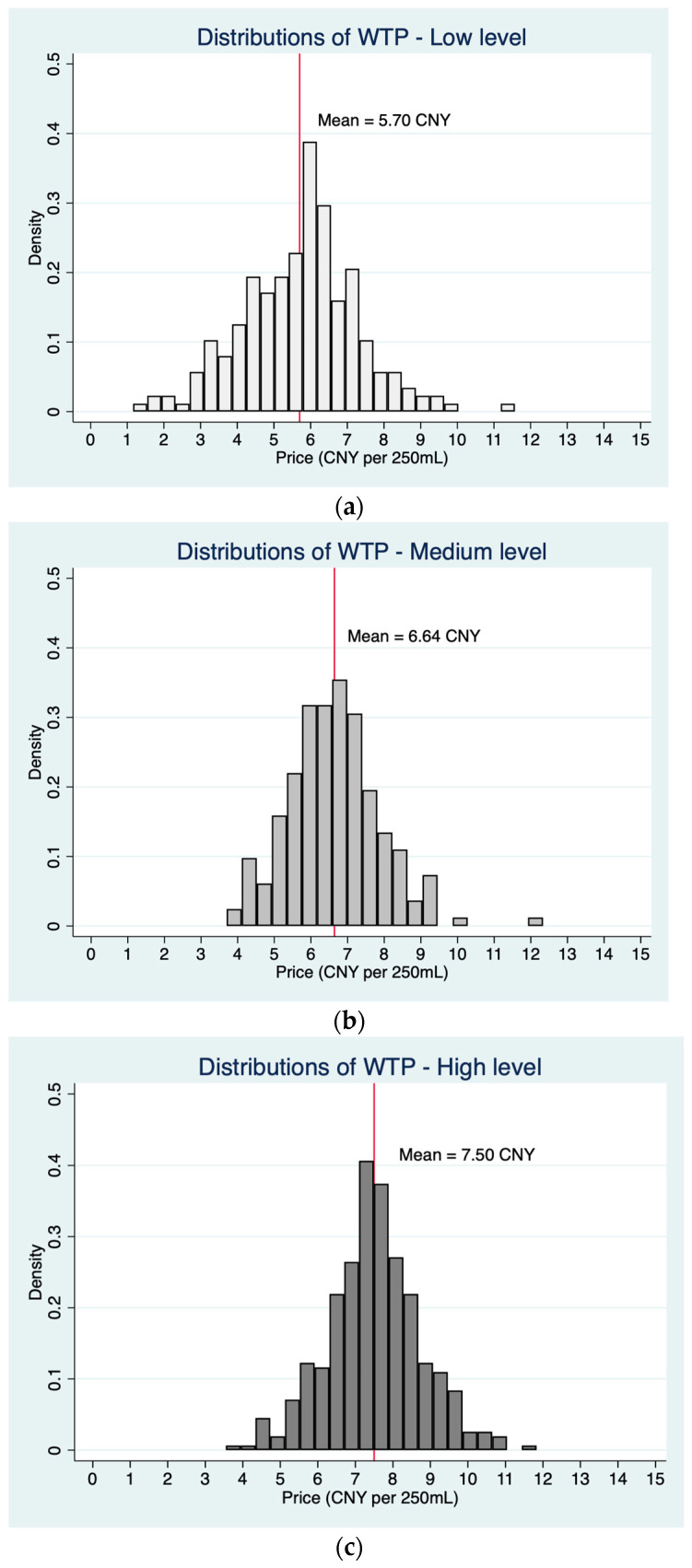
Distribution of willingness to pay (WTP) across different levels of environmental awareness. (**a**) Low level of environmental awareness. (**b**) Medium level of environmental awareness. (**c**) High level of environmental awareness.

**Figure 2 foods-13-00002-f002:**
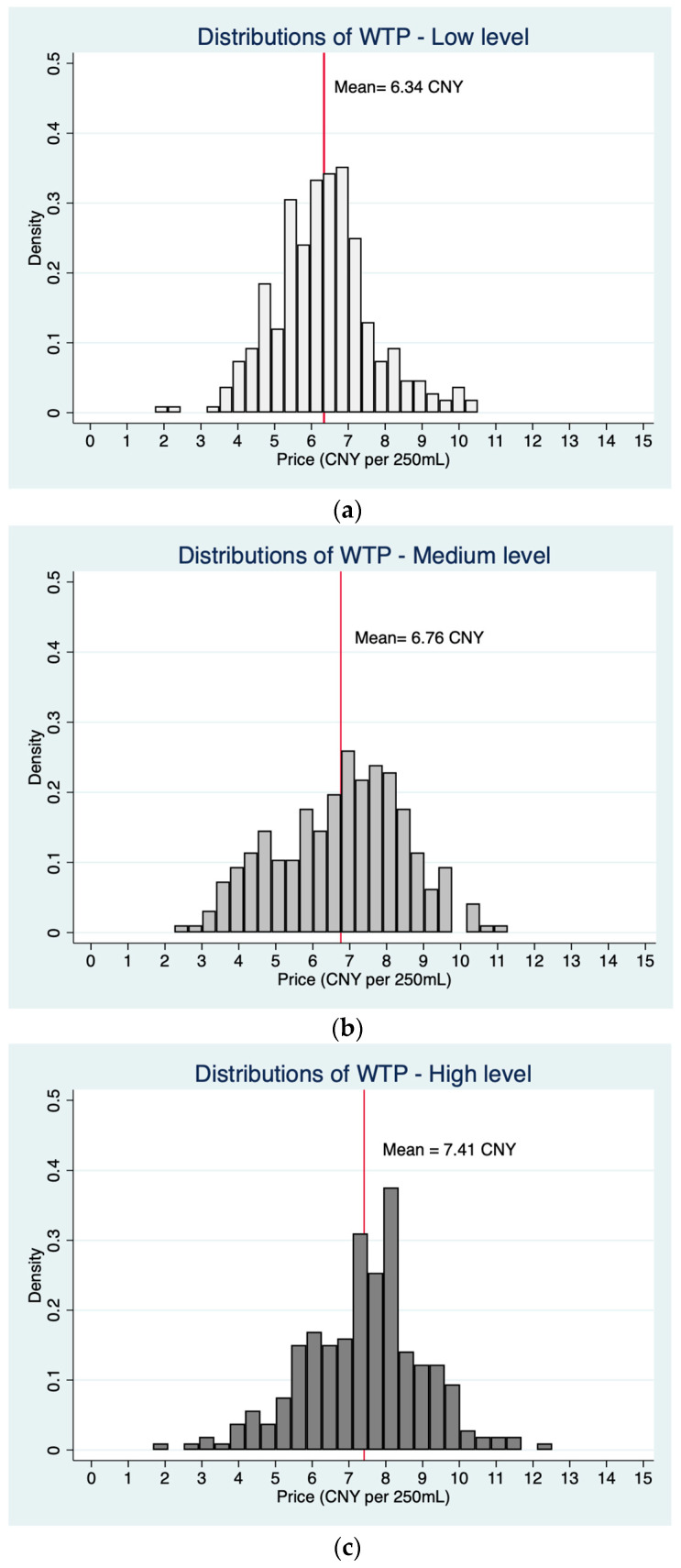
The distribution diagram of willingness to pay (WTP) based on three levels of health consciousness. (**a**) Low level of health consciousness. (**b**) Medium level of health consciousness. (**c**) High level of health consciousness.

**Figure 3 foods-13-00002-f003:**
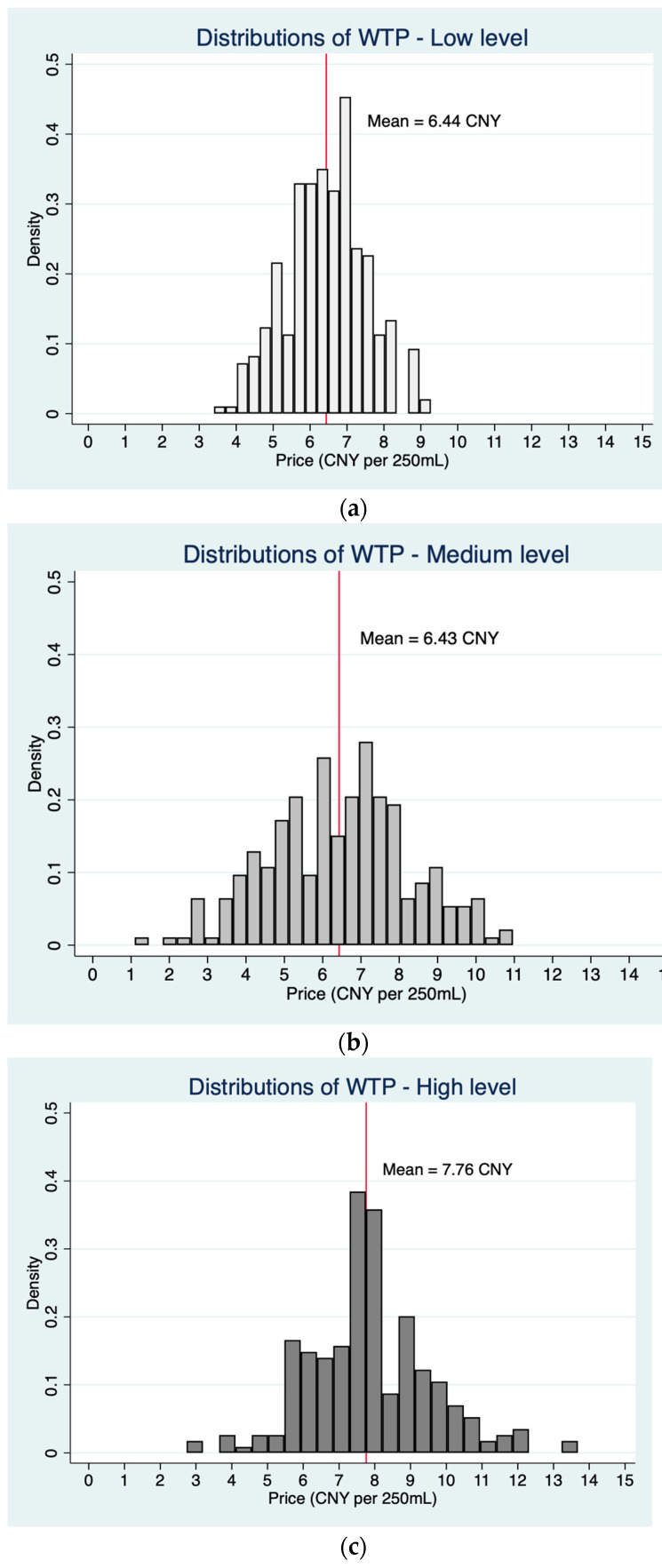
The distribution diagram of willingness to pay (WTP) based on three levels of food neophobia. (**a**) Low level of food neophobia. (**b**) Medium level of food neophobia. (**c**) High level of food neophobia.

**Table 1 foods-13-00002-t001:** Variables and definitions.

Variable	Definition	Value
Consumer characteristics		
Gender	Female = 0	487
	Male = 1	332
Age	Actual value of age	Mean = 32.4, St. Dev. = 11.215
Education level	Without a college degree = 0	83
	With a college degree = 1	736
Monthly income	Under CNY 3500 = 1	332
	CNY 3500–6000 = 2	216
	More than CNY 6000 = 3	271
Vegan	Vegan or vegetarian = 1	58
	I eat everything = 0	761
Health	Suffer from hyperglycemia (high blood sugar levels), hypertension (high blood pressure), or hyperlipidemia (high levels of lipids in the blood) = 1	75
	If not = 0	744
Lactose intolerant	Lactose intolerant = 1	74
	If not = 0	745
Urban residents	Living in urban area = 1	756
	Living in rural area = 0	63
Occupation	Engaged in health and environmental protection-related fields = 1	92
	Other occupations = 0	727
Relatives	My relatives and friends hardly purchase PBM = 1	241
	My relatives and friends occasionally purchase PBM = 2	499
	My relatives and friends frequently purchase PBM = 3	79
Product attributes ^a^		
Appearance	Degree of consumer preference for PBM appearance (color, packing method, unit packing capacity) of PBM	Mean = 3.56, St. Dev. = 0.98
Taste	Degree of consumer preference for sensory characteristics of PBM	Mean = 3.79, St. Dev. = 0.98
Convenience	Degree of consumer preference for purchase convenience of PBM	Mean = 3.86, St. Dev. = 0.97
Importance of price	Importance of price on purchase decisions for PBM	Mean = 3.98, St. Dev. = 0.93
Nutrition value	Degree of consumer preference for nutritional value of PBM	Mean = 4.40, St. Dev. = 0.89
Environmental benefits	Degree of consumer preference for environmental benefits of PBM in aspects of low carbon, animal welfare, conservation of ecological resources, and so on	Mean = 3.81, St. Dev. = 1.10

^a^ All the items were scored as follows: Very unimportant = 1; Unimportant = 2; Neither important nor unimportant = 3; Important = 4; Very important = 5.

**Table 2 foods-13-00002-t002:** Summary statistics of consumers’ environmental awareness, health consciousness, and food neophobia.

	Mean	Std. Dev.
Environmental awareness		
It is important to ensure adequate living space and respect the well-being of animals.	5.913	1.271
The production of food in an environmentally friendly manner is important.	5.977	1.175
It is important to utilize environmentally friendly packaging for food.	5.990	1.157
It is important to maintain ecological balance.	6.079	1.167
Health consciousness		
I frequently contemplate my well-being.	5.813	1.170
I place significant importance on maintaining my health.	6.094	1.118
I remain vigilant regarding any alterations in my health.	5.939	1.136
I typically stay mindful of my health.	5.839	1.158
I take personal accountability for my health.	5.960	1.162
I possess daily awareness of my health.	5.512	1.302
Food neophobia		
I frequently venture to taste novel and diverse food.	5.531	1.328
I lack trust in unfamiliar food.	4.486	1.681
I prefer not to consume food if its ingredients are unknown to me.	5.415	1.451
I have a fondness for food from various countries.	5.247	1.428
I feel unfamiliar food is peculiar.	4.849	1.600
I enjoy experiencing new dining establishments.	5.526	1.319

All the items were scored as follows: Strongly disagree = 1; Disagree = 2; Somewhat disagree = 3; General = 4; Somewhat agree = 5; Agree = 6; Strongly agree = 7.

**Table 3 foods-13-00002-t003:** Double-bounded dichotomy choice scheme.

Starting Price (CNY)	Second Price (CNY)	
	If Accepting the Starting Price	If Rejecting the Starting Price
7	8	6
8	9	7
9	10	8
10	11	9

**Table 4 foods-13-00002-t004:** Consumers’ willingness to pay (WTP) for plant-based milk (PBM).

First Price (CNY)	Accept	Percentage (%)	Second Price (CNY)	Accept	Percentage (%)
7	Yes	41.26	8	Yes	58.82
				No	41.18
	No	58.74	6	Yes	22.31
				No	77.69
8	Yes	36.73	9	Yes	68.06
				No	31.94
	No	63.27	7	Yes	13.71
				No	86.29
9	Yes	31.47	10	Yes	56.45
				No	43.55
	No	68.53	8	Yes	9.63
				No	90.37
10	Yes	27.73	11	Yes	57.38
				No	42.62
	No	72.27	9	Yes	13.21
				No	86.79

**Table 5 foods-13-00002-t005:** Preferences for product attributes and consumer characteristics that affect consumer WTP.

	WTP	Model 1	Model 2
Product attributes			
	Appearance	0.166 ***	−0.0157
		(0.023)	(0.209)
	Taste	0.113 ***	0.255
		(0.023)	(0.212)
	Convenience	−0.178 ***	0.555 ***
		(0.023)	(0.211)
	Importance of price	−0.229 ***	−1.087 ***
		(0.022)	(0.211)
	Nutrition value	0.161 ***	−0.171
		(0.023)	(0.215)
	Environment benefits	0.253 ***	0.320 *
		(0.019)	(0.183)
Consumer attitudes			
	Environmental awareness		0.522 **
			(0.211)
	Health consciousness		−0.158
			(0.220)
	Food neophobia		0.561 ***
			(0.202)
Consumer characteristics			
	Male	−0.642 ***	−0.547 *
		(0.032)	(0.299)
	Age	−0.01 ***	−0.00165
		(0.002)	(0.0139)
	Monthly income (CNY 3500–6000)	0.259 ***	0.837 *
		(0.002)	(0.473)
	Monthly income (above 6000)	0.411 ***	0.216
		(0.041)	(0.359)
	Education level (without a college degree)	1.137 ***	0.366
		(0.067)	(0.334)
	Urban residence	0.3 ***	0.543
		(0.067)	(0.559)
	Vegetarian	1.086 ***	0.565
		(0.067)	(0.526)
	Health	0.154 **	0.141
		(0.061)	(0.522)
	Lactose intolerance	0.244 ***	0.322
		(0.062)	(0.549)
	Occupation	1.016 ***	1.079 **
		(0.050)	(0.451)
	My relatives and friends occasionally purchase PBM	0.421 ***	0.690 **
		(0.034)	(0.323)
	My relatives and friends frequently purchase PBM	0.914 ***	0.140
		(0.066)	(0.533)
	Constant	6.23 ***	4.756 ***
		(0.066)	(1.060)

Standard errors in parentheses. *** *p* < 0.01, ** *p* < 0.05, * *p* < 0.1.

**Table 6 foods-13-00002-t006:** Regression analysis of willingness to pay (WTP) for plant-based milk across different levels of environmental awareness.

	(1)	(2)	(3)
Variables	Low Level of Environmental Awareness	Medium Level of Environmental Awareness	High Level of Environmental Awareness
Product attributes			
Appearance	0.709	−0.0425	−0.185
	(0.461)	(0.419)	(0.297)
Taste	−0.308	0.193	0.308
	(0.479)	(0.431)	(0.293)
Convenience	0.0120	1.153 ***	0.600 *
	(0.409)	(0.441)	(0.309)
Importance of price	−1.163 ***	−1.108 **	−1.103 ***
	(0.402)	(0.447)	(0.315)
Nutrition value	0.0476	−0.443	−0.0563
	(0.400)	(0.390)	(0.372)
Environmental benefits	0.127	0.0993	0.677 **
	(0.338)	(0.352)	(0.295)
Consumers’ characteristics			
Male	−0.697	−0.992 *	−0.324
	(0.618)	(0.568)	(0.441)
Age	0.0359	−0.00435	−0.00761
	(0.0305)	(0.0283)	(0.0194)
Monthly income (CNY 3500–6000)	−0.0965	0.128	0.419
	(0.765)	(0.683)	(0.525)
Monthly income (above CNY 6000)	−0.343	0.612	0.564
	(0.704)	(0.628)	(0.494)
Education level (without a college degree)	0.166	0.0384	−1.133 *
	(1.162)	(1.078)	(0.618)
Urban residence	0.364	−1.395	1.099
	(1.144)	(0.996)	(0.849)
Vegetarian	1.227	0.963	0.155
	(1.461)	(1.128)	(0.741)
Health	0.256	0.976	0.0708
	(1.006)	(1.150)	(0.761)
Lactose intolerance	−0.385	0.636	0.294
	(1.108)	(1.112)	(0.733)
Occupation	1.813 *	0.341	1.061 *
	(0.998)	(1.003)	(0.605)
My relatives and friends occasionally purchase PBM	1.034	0.365	0.879 *
	(0.708)	(0.610)	(0.485)
My relatives and friends frequently purchase PBM	0.259	0.225	0.0667
	(1.148)	(0.878)	(0.851)
Constant	6.235 **	8.686 ***	5.762 ***
	(2.464)	(2.307)	(1.936)
Observations	227	199	393

Standard errors in parentheses. *** *p* < 0.01, ** *p* < 0.05, * *p* < 0.1.

**Table 7 foods-13-00002-t007:** Regression analysis of willingness to pay (WTP) for plant-based milk across different levels of health consciousness.

	(1)	(2)	(3)
Variables	Low Level of Health Consciousness	Medium Level of Health Consciousness	High Level of Health Consciousness
Product attributes			
Appearance	0.749 **	−0.200	−0.371
	(0.356)	(0.366)	(0.373)
Taste	−0.356	0.581	0.360
	(0.348)	(0.394)	(0.380)
Convenience	−0.164	0.874 **	1.134 ***
	(0.314)	(0.443)	(0.410)
Importance of price	−1.026 ***	−1.289 ***	−1.160 ***
	(0.331)	(0.425)	(0.395)
Nutrition value	0.253	0.115	−0.585
	(0.315)	(0.441)	(0.445)
Environmental benefits	0.118	0.0819	0.972 ***
	(0.268)	(0.356)	(0.362)
Consumers’ characteristics			
Male	−0.157	−2.226 ***	0.0558
	(0.474)	(0.596)	(0.523)
Age	−0.0299	0.0452 *	−0.00609
	(0.0242)	(0.0270)	(0.0237)
Monthly income (CNY 3500–6000)	−0.190	0.417	0.455
	(0.583)	(0.676)	(0.636)
Monthly income (above CNY 6000)	−0.224	0.821	0.486
	(0.527)	(0.629)	(0.602)
Education level (without a college degree)	−0.993	−0.0131	−1.126
	(0.832)	(1.117)	(0.695)
Urban residence	−0.172	−0.0809	2.301 *
	(0.895)	(0.903)	(1.340)
Vegetarian	0.901	0.516	0.859
	(1.032)	(1.269)	(0.819)
Health	0.334	1.032	−0.453
	(0.996)	(1.007)	(0.843)
Lactose intolerance	−0.162	0.207	1.212
	(0.855)	(0.957)	(1.019)
Occupation	1.820 **	0.861	0.960
	(0.805)	(0.855)	(0.725)
My relatives and friends occasionally purchase PBM	0.904 *	0.667	0.710
	(0.506)	(0.616)	(0.595)
My relatives and friends frequently purchase PBM	0.282	0.386	0.202
	(0.819)	(1.018)	(1.036)
Constant	9.564 ***	4.571 *	4.049
	(1.997)	(2.605)	(2.483)
Observations	308	256	255

Standard errors in parentheses. *** *p* < 0.01, ** *p* < 0.05, * *p* < 0.1.

**Table 8 foods-13-00002-t008:** Regression analysis of willingness to pay (WTP) for plant-based milk across different levels of food neophobia.

	(1)	(2)	(3)
Variables	Low Level of Food Neophobia	Medium Level of Food Neophobia	High Level of Food Neophobia
Product attributes			
Appearance	0.117	−0.0387	−0.639
	(0.271)	(0.412)	(0.470)
Taste	−0.0445	0.298	1.308 **
	(0.278)	(0.386)	(0.521)
Convenience	0.534 *	−0.000926	1.071 **
	(0.280)	(0.374)	(0.488)
Importance of price	−1.104 ***	−1.254 ***	−1.182 **
	(0.288)	(0.371)	(0.490)
Nutrition value	0.218	−0.0377	−0.485
	(0.279)	(0.434)	(0.498)
Environmental benefits	−0.0902	0.958 ***	0.110
	(0.231)	(0.352)	(0.435)
Consumers’ characteristics			
Male	−0.555	−2.136 ***	−0.0632
	(0.421)	(0.639)	(0.558)
Age	0.0145	−0.0255	−0.00141
	(0.0192)	(0.0272)	(0.0267)
Monthly income (CNY 3500–6000)	−0.464	1.034	0.233
	(0.503)	(0.698)	(0.689)
Monthly income (above CNY 6000)	−0.210	0.778	0.592
	(0.478)	(0.634)	(0.654)
Education level (without a college degree)	0.190	−1.230	−1.257
	(0.734)	(0.915)	(0.827)
Vegetarian	0.708	−0.316	0.503
	(0.995)	(1.160)	(0.878)
Health	0.505	−0.253	0.692
	(0.747)	(1.131)	(0.972)
Lactose intolerance	−0.858	−0.0958	0.811
	(0.790)	(0.938)	(1.086)
Urban residence	−0.0898	−0.656	1.691
	(0.735)	(1.044)	(1.131)
Occupation	0.416	0.0356	2.337 ***
	(0.715)	(0.824)	(0.864)
My relatives and friends occasionally purchase PBM	0.983 **	0.951	−0.0741
	(0.444)	(0.613)	(0.638)
My relatives and friends frequently purchase PBM	0.456	1.355	−1.871
	(0.749)	(0.918)	(1.170)
Constant	7.130 ***	8.994 ***	6.330 ***
	(1.714)	(2.489)	(2.254)
Observations	314	254	251

Standard errors in parentheses. *** *p* < 0.01, ** *p* < 0.05, * *p* < 0.1.

## Data Availability

The datasets are available from the corresponding author on reasonable request.

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
