# Peer review of "Consumers’ Preferences and Attitudes towards Plant-Based Milk"

_foods, 2023, doi:10.3390/foods13010002_

Round 1

Reviewer 1 Report

Comments and Suggestions for Authors

This is a good quality submission. The research problem is relevant and important. The applied research method is appropriate. The results are presented clearly.

Please explain in the manuscript how the target sample size was determined.

Please compare your sample characteristics with the general population.

Please write a section about managerial and policy implications.

Please write a section about future research directions.

Minor corrections:

line 218 - include

267 - have

279 - engaged

322 - level have

427 - [65]

460 - take into account

Comments on the Quality of English Language

acceptable

Author Response

Consumers Preferences and Attitudes Towards Plant-Based Milk

Response to Reviewer 1 Comments

1Summary

Thank you very much for taking the time to review this manuscript.  Please see the attachment and the corresponding revisions in track changes in the re-submitted files.

2Questions for General Evaluation

Reviewer’s Evaluation

Response and Revisions

Does the introduction provide sufficient background and include all relevant references?

Yes

Thank you!

Are all the cited references relevant to the research?

Yes

Thank you!

Is the research design appropriate?

Yes

Thank you!

Are the methods adequately described?

Must be improved

We appreciate the reviewer' s feedback on the description of our methods. We understand the need for clarity and thoroughness in this section. To address this,  we have expanded the Methodology section and provided a more detailed explanation of our experimental design,  data collection and analysis procedures. We believe these changes will enhance the reader' s understanding of our methods and the reproducibility of our study. 

Are the results clearly presented?

Can be improved

Thank you for your suggestion to improve clarity in the presentation of our results. We have revised this section to provide a more concise and clear presentation of our findings. 

Are the conclusions supported by the results?

Can be improved

We appreciate your feedback regarding the coherence between our results and conclusions. We have reevaluated our conclusions in light of our results,  and adjusted them to more accurately reflect our findings. We have also strengthened our discussion of the implications of our results,  ensuring they are backed by the data presented. 

3Point-by-point response to Comments and Suggestions for Authors

Comments 1:This is a good quality submissionThe research problem is relevant and importantThe applied research method is appropriateThe results are presented clearly.   Please explain in the manuscript how the target sample size was determined

Response 1: Thank you for pointing this out.  We agree with this comment.  Therefore,  we have explained how we determined the target sample size.  Here is updated text in our manuscript:

We sent out a total of 900 questionnaires using the online survey platform Wenjuanxing (https://www.  wjx.  cn),  with 300 questionnaires in each region. We obtained 819 valid responses.  To ensure the quality of the survey,  we set a minimum response time of 3 seconds per question to prevent participants from rushing through the questions. Additionally,  we implemented technical identification measures, including IP and device tracking,  to prevent duplicate responses from the same user and avoid repeated sampling. Participants who do not consume milk or milk alternatives were also included in the survey.  Such participants, representing a potential market for PBM,  may offer unique insights into the barriers to consumption of PBM products. 

Comments 2:Please compare your sample characteristics with the general population

Response 2: Agree. We need to compare your sample characteristics with the general population.  We updated the second paragraph in section 4:

The study sample consisted of 59% female and 41% male. The average age of participants is 32.  4 years.  A majority of the participants hold a college degree, indicating a higher level of education compared to the broader Chinese population. The income levels of participants are fairly distributed across three categories. Dietary preferences show that 7% of participants adhere to a vegan or vegetarian diet and 9%report lactose intolerance. In this study, we conducted a survey of consumer preferences for six attributes of PBM by the Likert scale,  including appearance,  taste,  nutritional value,  environmental benefits,  price,  and purchase convenience.  The basic information about PBM such as its nutritional health and environmental benefits were provided. 

Comments 3:Please write a section about managerial and policy implications

Response 3: Agree. 

Comments 4:Please write a section about future research directions

Response 4: Agree. Thank you for your insightful suggestion. We agree that discussing future research directions would indeed strengthen our paper.  Here is a proposed section(on Page 21)on this matter:

Longitudinal studies could track changes in consumer attitudes over time.  It would also be beneficial to compare stated preferences and willingness to pay with actual purchasing behavior to reinforce the findings. As alternative proteins evolve, exploring consumer responses to options like insect-based proteins or lab-grown meat could provide valuable insights into the future protein market. Additionally,  understanding the influence of specific information about PBM' s health and environmental benefits on consumer perception could be instrumental in shaping educational and marketing strategies.  

4Response to Comments on the Quality of English Language

Point 1:acceptable

Thank you!

5Additional clarifications

Minor corrections:

line 218-include

267-have

279-engaged

322-level have

427-[65]

460-take into account

Reviewer 2 Report

Comments and Suggestions for Authors

The article "Consumers' Preferences and Attitudes Towards Plant-Based Milk" is interesting and focuses on current topics but contains many ambiguities that need clarification. Is the problem discussed globally, or does it concern a selected world region?

The work appears to be a review rather than a research work. The authors should, therefore, redact the manuscript so that the information contained therein does not raise any doubts.

Introduction.

The purpose of the work should be reworded. In its current form, it is not understandable. One can only guess the purpose of the manuscript, taking into account the description of the goal in the article abstract.

Lines: 62-169.

According to the reviewer, the review is general; the information is

presented chaotically and does not form a coherent whole. Therefore, it needs to be clarified what the authors wanted to convey. How does the information in the review connect to the topic and purpose of the work?

It is, therefore, necessary to organize the messages and compare them in terms of the analyzed problem.

Lines 233-234. Where did the authors send the questionnaires? To whom (what group of respondents, how were they selected (was there a key?)).

I propose to add a chapter: Materials and Methods. Additionally, specific subsections would undoubtedly be helpful.

It isn't easy to understand and extract information from the totality that will allow you to understand what the authors wanted to present. There needs to be more logical continuity.

Some results are even before the results section (line 305).

Lines 307-309. The authors state: “In our sample, the mean value of WTP for PBM was found to be 7.919 CNY per 250mL with a standard deviation of 0.979 (Table 5), and the density of WTP followed a normal distribution.” Please pay attention to the way the results are presented. Is the presented fragment understandable? The reviewer suggests using abbreviations where necessary. Abbreviations should certainly not start the Results section. The note applies to the entire work.

Is it necessary to describe the standard deviation?

Did the authors describe all the methods used? What about the statistical analysis of the results?

Please eliminate duplication of results in charts and tables.

Author Response

Consumers Preferences and Attitudes Towards Plant-Based Milk

Response to Reviewer 2 Comments

1.Summary

Thank you very much for taking the time to review this manuscript. Please find the detailed responses below and the corresponding revisions in the re-submitted files.

2.Questions for General Evaluation

Reviewer’s Evaluation

Response and Revisions

Does the introduction provide sufficient background and include all relevant references?

Must be improved

Response:We appreciate your feedback regarding the introduction.We will revise this section to provide more background information on the topic and ensure that all relevant references have been included.This will help to better establish the context and significance of our study.

Revisions: We will expand on the current trends and research in the field of plant-based milk(PBM)and consumer behavior. We' ll also make sure to include a wider range of references that are relevant to our study.

Are all the cited references relevant to the research?

Not applicable

Thank you for your evaluation.We are committed to maintaining academic rigor and will continue to ensure that all references cited in our manuscript are pertinent to our research.

Is the research design appropriate?

Must be improved

Response: Thank you for your comments on the research design. We acknowledge that more detail could be provided to better illustrate our design and its appropriateness for the research question at hand.

Revisions: We will revise the" Survey" section(refer to" Materials and Methods" )to provide a more comprehensive description of our research design,including the choice of the study population, sampling strategy,and rationale behind the methods employed.

Are the methods adequately described?

Must be improved

Response: We appreciate your feedback on the description of our methods. We understand that a more detailed explanation of our data collection and analysis methods would contribute to the transparency and replicability of our study.

Revisions: We will expand the" Survey" section to provide a more detailed description of the double-bounded dichotomy choice(DBDC)method used to calculate consumers'  willingness to pay(WTP)for PBM, and the statistical analysis of the results.

Are the results clearly presented?

Must be improved

Response:We acknowledge your concerns about the clarity of our results presentation. We understand that the use of abbreviations and the structure of our Results section may have led to confusion.

Revisions: We will revise the Results section to ensure a clear and logical presentation of our findings. We will clearly define all abbreviations at their first occurrence and ensure that results are not prematurely presented before this section.

Are the conclusions supported by the results?

Must be improved

Response:We appreciate your feedback on the connection between our results and conclusions.We understand the importance of ensuring that our conclusions directly stem from our results.

Revisions: We will revise the Conclusions section to more explicitly link our findings to our conclusions.We will also ensure that our conclusions are supported by the results presented in the body of the paper.

3.Point-by-point response to Comments and Suggestions for Authors

Comments 1:The article" Consumers'  Preferences and Attitudes Towards Plant-Based Milk" is interesting and focuses on current topics but contains many ambiguities that need clarification.Is the problem discussed globally,or does it concern a selected world region? The work appears to be a review rather than a research work.The authors should,therefore,redact the manuscript so that the information contained therein does not raise any doubts.

Response 1: Thank you for your valuable feedback and constructive criticism.We apologize if our scope was not explicitly clear in the manuscript. In response to your comment, this study primarily focuses on Chinese consumers, a significant and somewhat under-researched demographic in the context of plant-based milk consumer behavior.We chose to focus on this group due to their increasing consumption of plant-based milk and the potential differences in their motives and attitudes compared to their counterparts in Europe and North America, which are regions more extensively covered in existing literature.

In light of your comments, we will revise the manuscript to more clearly define the scope and focus of our research. Additionally, while our study does incorporate a review of existing literature,it primarily constitutes original research characterized by the survey and analysis of Chinese consumers'  preferences and attitudes. We will work to emphasize this point more clearly in our revision to avoid any confusion. Again,we appreciate your feedback and look forward to improving the clarity and precision of our manuscript.

We added this sentence to the last paragraph of the introduction:

While our study references global trends,its primary focus lies on exploring the preferences and attitudes of Chinese consumers towards plant-based milk, an important yet under-researched demographic in this growing market sector.

Comments 2:Introduction. The purpose of the work should be reworded.In its current form,it is not understandable. One can only guess the purpose of the manuscript,taking into account the description of the goal in the article abstract. Lines:62-169. According to the reviewer,the review is general;the information is presented chaotically and does not form a coherent whole.Therefore,it needs to be clarified what the authors wanted to convey. How does the information in the review connect to the topic and purpose of the work?  It is, therefore,necessary to organize the messages and compare them in terms of the analyzed problem.

Response 2:Agree.Thank you for your valuable comments. We understand your concerns about the clarity of the work' s purpose and the organization of the review. In response to your comments,we will revise the introduction to state the purpose of our study more explicitly. We agree that the current phrasing may lead to ambiguity, and we appreciate your suggestion to improve this aspect.

Regarding the review section, we acknowledge that the information could be more coherently organized to better demonstrate its relevance to consumers' preferences and attitudes towards plant-based milk.The first segment of our literature review focuses on the substantial growth of the Plant-Based Milk (PBM) market, both globally and regionally, driven by factors such as health consciousness, environmental awareness, dietary restrictions,cultural acceptance, and industry innovation. The second segment seeks to provide a comprehensive understanding of consumer attitudes towards PBM, highlighting the influential roles of environmental awareness,health consciousness, and food neophobia. The third segment delves into consumers' willingness to pay (WTP) for PBM, discussing the utility and potential shortcomings of the Double-Bounded Dichotomous Choice Contingent Valuation Method (DBDC-CVM) for eliciting WTP values. We acknowledge, however, the need for a more structured presentation to better connect previous research with our study. We will revise the Literature review section to more logically present the information and explicitly link each reviewed study to the research questions and objectives of our work.

We modify the abstract to make the purpose of our study clear:

This study aims to identify the main factors that affect consumer preferences and attitudes towards PBM,and examine the effect of consumer attitudes,including environmental awareness,health consciousness and food neophobia on WTP.

Comments 3:Lines 233-234. Where did the authors send the questionnaires?To whom(what group of respondents,how were they selected(was there a key?)).

Response 3: Thank you for your insightful comment. Our survey was conducted using the Wenjuanxing online survey platform (https://www.wjx.cn) within the Jing-Jin-Ji metropolitan region, encompassing the municipalities of Beijing and Tianjin, and Hebei Province in North China. This region represents the largest urbanized megalopolis area in Northern China. We disseminated a total of 900 questionnaires, divided evenly across the three regions,resulting in 819 valid responses.

In response to your query about our respondent demographics, we have made the following changes in the manuscript:

The study sample consisted of 59%female and 41%male.The average age of participants is 32.4 years.A majority of the participants hold a college degree,indicating a higher level of education compared to the broader Chinese population.The income levels of participants are fairly distributed across three categories(under 3500 CNY,3500-6000 CNY,More than 6000 CNY).

We hope this additional information provides a clearer understanding of the demographics of our respondents and enhances the context of our study.

Comments 4: I propose to add a chapter: Materials and Methods. Additionally, specific subsections would undoubtedly be helpful. It isn' t easy to understand and extract information from the totality that will allow you to understand what the authors wanted to present.There needs to be more logical continuity. Some results are even before the results section(line 305).

Response 4:Thank you for your helpful suggestion.We understand your point about the potential benefits of having a distinct " Materials and Methods" chapter,along with specific subsections for improved clarity and continuity.

In our manuscript,we have incorporated a section named "Survey", which indeed serves a similar purpose to a traditional "Materials and Methods" section. In this section“Survey”, we primarily detail the origins of our survey data and describe how we utilized the conditional valuation method to obtain data on consumers' willingness to pay. We appreciate your feedback and recognize that this might have been unclear due to our unconventional naming. We will consider renaming this section to "Materials and Methods" for clarity and adherence to common academic conventions.

Regarding the continuity of the paper, we acknowledge your concerns and will revisit the structure of the manuscript to ensure logical flow and coherence. We also note your observation about some results appearing before the designated "Results" section. This was unintentional, and we will rectify this to ensure that all results are presented in the appropriate section.

We are grateful for your time and expertise in reviewing our paper,and we appreciate your insightful comments that will undoubtedly aid in improving our manuscript.

Comments 5: Lines 307-309. The authors state:“In our sample,the mean value of WTP for PBM was found to be 7.919 CNY per 250mL with a standard deviation of 0.979(Table 5), and the density of WTP followed a normal distribution.”Please pay attention to the way the results are presented. Is the presented fragment understandable?The reviewer suggests using abbreviations where necessary. Abbreviations should certainly not start the Results section.The note applies to the entire work.

Response 5: We appreciate your feedback regarding the presentation of our results and the use of abbreviations.We understand that the use of abbreviations,especially at the beginning of the Results section, might lead to confusion.

We make changes of the title in section 5.1 and 5.2: 5.1.Willingness to pay(WTP)for plant-based milk(PBM)and its Influencing Factors & 5.2.The Effect of Consumer Attitudes on willingness to pay(WTP).

We also revised the mentioned lines to read:In our sample,the mean value of willingness to pay(WTP)for plant-based milk(PBM)was found to be 7.919 CNY per 250mL with a standard deviation of 0.979(Table 5),and the density of WTP followed a normal distribution.

We will also ensure that all abbreviations are properly defined at their first occurrence in the text, not just in the Results section, but throughout the entire manuscript. We hope these revisions will increase the clarity and readability of our work.

Comments 6:Is it necessary to describe the standard deviation?

Response 6: Thank you for your question. The inclusion of the standard deviation in scientific or research reports is generally considered important as it provides a measure of how spread out the values are in a data set. In our case, it helps to understand the variability in the willingness to pay (WTP) among the survey respondents. That being said, if the focus of our report is more on the average value of WTP and less on the variability, the standard deviation might be less of interest to our readers.We would be glad to revise our manuscript based on your suggestion and the intended focus of our study. As always, we appreciate your feedback and look forward to any additional comments or suggestions you may have to help us improve our work.

Comments 7:Did the authors describe all the methods used?What about the statistical analysis of the results?

Response 7:Thank you for your question. We appreciate your attention to detail and the question posed.In our study, we indeed employed the double-bounded dichotomy choice (DBDC) method to calculate consumers' willingness to pay(WTP)for Plant-Based Milk(PBM).We also conducted a regression analysis to examine the effect of consumer attitudes,including environmental awareness,health consciousness and food neophobia on WTP. However,we acknowledge that we may not have elaborated sufficiently on the specific statistical methods used in our analysis.To rectify this,we will add more details about the statistical procedures we used.

To clarify,Table 4 presents the statistical analysis of consumers' WTP for PBM,and Table 5 provides a regression analysis of the product attributes and consumer characteristics that affect consumer WTP.Tables 6 through 8 present the results of regression analyses,which examine the willingness to pay(WTP)for plant-based milk in relation to varying degrees of environmental awareness,health consciousness,and food neophobia,respectively. We hope this clarifies our methods and thank you for your suggestion.We will ensure to provide a more comprehensive description of our statistical analysis in the revised manuscript.

Comments 8:Please eliminate duplication of results in charts and tables.

Response 8:Thank you for your valuable feedback about the duplication of results in charts and tables.We understand that this could potentially lead to redundancy and confusion. Based on your comment,we have revisited Tables 6 through 8 and made necessary adjustments to avoid overlapping information.We have also revised the notes for these tables to clarify the data presented and ensure they are more concise and easier to understand.We appreciate your attention to detail and the time you have taken to review our manuscript.Your comments are instrumental in improving the quality of our work.

Reviewer 3 Report

Comments and Suggestions for Authors

I have completed my review of Consumers' Preferences and Attitudes Towards Plant-Based Milk.  The article the impact of different attitudes, particularly environmental awareness, food neophobia, and health consciousness, on willingness-to-pay of plant-based milk.

The contribution of the paper is relatively minor.  This topic has been studied before and the findings of the paper are broadly in line with the literature.  The paper does survey Chinese consumers, who are major purchasers of plant-based milk and could potentially have different motives than European and North American consumers (who are more widely studied).

I have two major comments for the authors.  The first is to include all the explanatory variables in a single model, rather than estimate different models for different subgroups.  The second is to improve the literature review by clearly delineating what papers reference plant-based milk and which ones refer to other products (this was unclear and sometimes incorrect in their paper).

Best,

Comments on the Quality of English Language

The paper was generally well-written and easy to read.  However, there were several grammatical errors.  Most of these would be handled in a technical edit.  However, the authors should carefully read through their paper before re-submitting.

Author Response

Consumers Preferences and Attitudes Towards Plant-Based Milk

Response to Reviewer 3 Comments

1.Summary

Thank you very much for taking the time to review this manuscript. We appreciate the time and effort you have taken to review our work. Please find our responses to each of your comments below.

2.Questions for General Evaluation

Reviewer’s Evaluation

Response and Revisions

Does the introduction provide sufficient background and include all relevant references?

Can be improved

We appreciate your positive feedback regarding the research design. We aimed to ensure that our design effectively addressed the research question, and your comment confirms that we have been successful in this regard.

Are all the cited references relevant to the research?

Can be improved

We appreciate your valuable feedback. In response to your comments, we have meticulously restructured our references and rectified some discrepancies. Moreover, we have incorporated the four pivotal papers you recommended. We are confident that these additions will significantly enhance the robustness of our research. Thank you for your constructive suggestions.

Is the research design appropriate?

Yes

We are grateful for your positive feedback.

Are the methods adequately described?

Can be improved

Thank you for your feedback. We understand the importance of clearly describing the research methods to ensure transparency and reproducibility. In response to your comment, we have revised this section to provide more details about our methodology, including the specific procedures followed, the rationale behind our choices, and any potential limitations.

Are the results clearly presented?

Can be improved

We appreciate your feedback on the presentation of our results. We understand the necessity of clearly and comprehensively presenting our findings.  We have revised the results section to improve clarity, focusing on ensuring that our descriptions are precise and that all necessary data is included.

Revision: We add a new model including all explanatory variables, refer to model 2 in Table 5.

Are the conclusions supported by the results?

Yes

We are grateful for your positive feedback regarding the alignment of our conclusions with our results. We endeavored to draw conclusions that are directly supported by our findings, and your comment confirms that we have achieved this aim.

3.Point-by-point response to Comments and Suggestions for Authors

Comments 1: I have completed my review of Consumers' Preferences and Attitudes Towards Plant-Based Milk. The article the impact of different attitudes, particularly environmental awareness,food neophobia,and health consciousness,on willingness-to-pay of plant-based milk. The contribution of the paper is relatively minor. This topic has been studied before and the findings of the paper are broadly in line with the literature. The paper does survey Chinese consumers,who are major purchasers of plant-based milk and could potentially have different motives than European and North American consumers(who are more widely studied). I have two major comments for the authors. The first is to include all the explanatory variables in a single model, rather than estimate different models for different subgroups.

Response 1: Thank you for pointing this out. We agree with this comment. Therefore, we have now included a new model that integrates all the explanatory variables. to ensure a comprehensive understanding of the factors influencing the willingness to pay (WTP) for Plant-Based Milk (PBM). This approach allows us to assess the cumulative effect of product attributes, consumer attitudes, and consumer characteristics on WTP. You can refer to Model 2 in Table 5, where we have presented the results of this comprehensive regression model.

Comments 2: The second is to improve the literature review by clearly delineating what papers reference plant-based milk and which ones refer to other products(this was unclear and sometimes incorrect in their paper)

Response 2: Agree. We appreciate your valuable feedback. In response to your comments, we have meticulously restructured our references and rectified some discrepancies. Moreover, we have incorporated the four pivotal papers you recommended. We are confident that these additions will significantly enhance the robustness of our research. Thank you for your constructive suggestions.

Comments 3: I enjoyed reading through this article. I offer some suggestions for improvement below. It is not clear to me why the authors run separate models for individuals with differing levels of environmental awareness,health consciousness,and food neophobia. A more straightforward and standard approach would be to include these measures as explanatory variables in the primary regression model. I also find the authors interpretation of differences between groups to be off. For example,they state that "Consumers who exhibited high food neophobia tend to prioritize taste as well as purchase convenience." (line 385). This is not quite true. The results do not show that food neophobic consumers are. more likely to value taste, instead they show that those who do value taste are have a higher WTP for PBM. I think the interpretation would be simpler if food neophobia (and the environmental and health variables) were simply included in the primary model. If the authors believe that the effect of product attributes on WTP are moderated by environmental awareness, health consciousness, or food neophobia, then they could include interaction terms in the regression equation.

Response 3: Agree. Thank you for taking the time to review our manuscript and for your insightful comments. We appreciate your feedback on the use of separate models for individuals with different levels of environmental awareness, health consciousness, and food neophobia. We understand why you propose it. However, we chose to run separate models for two main reasons. One is to simplify interpretation. We found that separate models provided a clearer perspective on the distinct effects of product attributes on WTP across varying levels of environmental awareness, health consciousness, and food neophobia. While interaction terms could offer similar insights, they tend to complicate the interpretation of results. The other is to avoid multicollinearity. Our model includes six core explanatory variables and three potential moderators. Incorporating interaction terms would result in an additional 18 variables, which could potentially lead to multicollinearity. This, in turn, could inflate the variance of the regression coefficients, thus destabilizing results and complicating interpretation. In response to your feedback, we have now added a new model (Model 2 in Table 5) that integrates all the explanatory variables, including the three moderators (environmental awareness, health consciousness, and food neophobia).

Regarding the interpretation issue you've pointed out, we agree with your observation. In model (3) in Table 8, the coefficient for taste (1.308) and convenience(1.071) is significant. It indicate that consumers  with high level of food neophobia, who attach importance of product taste and purchase convenience will have higher WTP for PBM. On the other hand, the coefficients for taste and convenience are not significant in Models (1) and (2) in Table 8. This implies that, for consumer groups with low and medium levels of food neophobia, the preference for product taste and purchase convenience does not significantly influence their WTP for PBM.

We have updated the phrasing of this observation in section 5.2.3 as follows:  For consumers exhibiting a high level of food neophobia, those who attach importance of product taste and purchase convenience will have higher WTP for PBM.

Thank you once again for your constructive feedback. It has certainly helped us improve the robustness and clarity of our work.

Comments 4:There needs to greater clarity about what products the authors are referring to in their literature review.For example:

a.“This perception greatly influences the readiness to replace dairy milk with PBM[30],underlining the central function of health consciousness in driving consumer choices”(line 121).I believe the reference in question examines the literature for plant-based meat,not milk.

b.“In the realm of PBM,Van Loo et al.[38]conducted one of the earliest studies examining consumer WTP.They found that consumers were willing to pay a premium for PBM due to its perceived health and environmental benefits.”(line 150).I believe this reference is to a study of organic chicken breast.Similarly,reference 39 seems to deal with chicken as opposed to milk

c.More broadly,the authors should clearly delineate when a study has examined WTP for(a)plant-based milk,(b)another plant-based product(usually meat),or(c)another product entirely.

Response 4: Thank you for your careful reading of our manuscript and for pointing out areas where we need to provide more clarity regarding the products discussed in our literature review. We appreciate your suggestion to clearly delineate when a study has examined WTP for (a) plant-based milk, (b) another plant-based product (usually meat), or (c) another product entirely. You are correct in pointing out that certain references do not directly pertain to plant-based milk (PBM), and we apologize for any confusion caused. The intention behind including these references was to highlight general trends in consumer behavior towards plant-based or environmentally friendly products. However, we understand that this could lead to confusion and will endeavor to make the distinctions clearer in our revised manuscript.

a) For the reference [30] you mentioned, we will clarify that this study pertains to plant-based meat, not milk.

b) Similarly for references [38] and [39], we will specify that these studies relate to organic chicken breast and not PBM. We will clarify that we intended to highlight their findings regarding consumers' willingness to pay a premium for products perceived to have health and environmental benefits, a sentiment we hypothesize is applicable to PBM.

c) Moreover, we will make sure to explicitly mention the specific product examined in each study we cite in our literature review section. This will help to provide a clearer context for our study, which focuses on WTP for PBM.

Once again, we appreciate your valuable feedback and will make the necessary adjustments to improve the clarity of our manuscript.

Comments 5: Some additional papers that the authors should cite in their lit review and compare their results with:

Slade,Peter,and Mila Markevych."Killing the sacred dairy cow?Consumer preferences for plant-based milk alternatives."Agribusiness(2023).

Summary:A discrete choice survey examining preferences for plant-based milk.

Rombach,Meike,David L.Dean,and Christopher Gan."“Soy Boy vs.Holy Cow”—Understanding the Key Factors Determining US Consumers’Preferences and Commitment to Plant-Based Milk Alternatives."Sustainability 15.18(2023):13715.

Summary:A survey exploring factors that explain preferences for plant-based milk.

Possibly,Favour Esene’s MSc thesis(University of Kentucky)which also measures WTP for dairy and non-dairy milk

Slade,Peter."Does plant-based milk reduce sales of dairy milk?Evidence from the almond milk craze."Agricultural and Resource Economics Review 52.1(2023):112-131.1

Summary:models consumer demand for plant-based milk,capturing some sociodemographic differences between households who consume and don’t consume dairy milk.

Dharmasena,Senarath,and Oral Capps."Unraveling demand for dairy-alternative beverages in the United States:The case of soymilk."Agricultural and Resource Economics Review 43.1(2014):140-157.

Summary:analyzes sociodemographic differences between soy and dairy milk consumers,though perhaps a bit dated.

Response 5:Thank you for the recommended references and the detailed summaries. These papers indeed provide valuable insights that align with our research topic, and we appreciate your taking the time to bring them to our attention. These studies will undoubtedly enrich our literature review and offer additional perspectives for comparing and discussing our results. I will diligently review these literature and integrate them into my manuscript. I am confident that these resources will significantly strengthen the depth and scope of my research. Thank you once again for offering these valuable and pivotal references.

Comments 6: I would find it easier to understand the empirical model if the authors explained it after discussing their experiment (i.e.,change the order of section 3 and 4).

Response 6:Thank you for sharing your thoughts regarding the structure of our paper. We understand that shifting the order of the sections may improve the flow and comprehension of our study. Your suggestion to explain the empirical model after discussing the experiment seems quite reasonable. We will, therefore, revise the manuscript to discuss the experiment details first in Section 3 and then proceed to the explanation of the empirical model in Section 4. We appreciate your thoughtful comments, and we believe that this change will enhance the reader's understanding and improve the overall flow of the paper.

Comments 7: “The maximum and minimum vale of WTP estimation were 4.827 and 11.321”(line 309)I think it is important to explain that these are expected WTP estimates(i.e.,ziβ), with the only variation coming from differences in zi. Actual WTP likely varies considerably more. A similar point can be made about all the WTP graphs.I am also curious why the graphs show(expected?)WTP values below 4.827.

Response 7: Thank you for your detailed review and insightful comments. You are correct in your observation that the values 4.827 and 11.321 we reported are indeed the expected WTP estimates (i.e., ziβ), accounting only for differences in zi. As you pointed out, individual WTP values may show wider variation due to unobserved heterogeneity among respondents and other factors not included in zi. We appreciate your suggestion to clarify this point in our manuscript. In the revised version, we will explicitly state that these are expected WTP values that reflect variations in zi and that actual individual WTP may vary more substantially around these estimates. As for the WTP graphs, the observed values below 4.827 could be due to different individual-specific random effects or error terms, which are not captured by the expected WTP (ziβ). We apologize for the confusion. Your comments have helped us realize areas where we need to provide more clarity, and we will ensure that these points are addressed in our revised version. Thank you again for your insightful feedback. We add further explanation of the willingness to pay model in the section 4 (Analytical framework): This is the expected WTP values that reflect variations in zi and that actual individual WTP may vary more substantially around these estimates.

In 5.2.1 section, we added this sentence: The observed values below 4.827 could be due to different individual-specific random effects or error terms, which are not captured by the expected WTP (ziβ).

Comments 8: The paper was generally well-written and easy to read. However, there were several grammatical errors. Most of these would be handled in a technical edit. However, the authors should carefully read through their paper before re-submitting.

Response 8:Thank you for your constructive feedback on our manuscript. We appreciate your positive comments about the overall readability of our paper. We recognize the importance of clear and error-free communication in presenting our research. We regret that several grammatical errors have slipped through. To enhance the quality of our manuscript, we will meticulously review and correct these issues. We also agree with your suggestion about a technical edit, and we will look into engaging a professional editing service to ensure the linguistic accuracy of our paper. We are grateful for your careful review and valuable remarks. They will undeniably assist us in improving the quality of our work. We look forward to presenting a thoroughly revised and polished manuscript upon resubmission.

4.Response to Comments on the Quality of English Language

Point: The paper was generally well-written and easy to read. However,there were several grammatical errors. Most of these would be handled in a technical edit. However, the authors should carefully read through their paper before re-submitting.

Response: We appreciate your positive feedback on the overall readability of our paper. We regret the presence of several grammatical errors and acknowledge the importance of careful proofreading in maintaining the quality of our manuscript. In response to your comments, we will thoroughly review the entire manuscript again and correct any grammatical errors. We will also consider engaging a technical editor to ensure the language of our paper meets academic standards. Thank you for pointing out this important aspect.We are committed to improving the quality of the presentation and ensuring that our paper is error-free before re-submission.Your feedback is greatly appreciated.

Round 2

Reviewer 2 Report

Comments and Suggestions for Authors

I want to thank the authors for the changes introduced.

The work still requires changes because there are questionable parts.

Lines 57-58. Incomprehensible statement: "an important yet under-re-searched demographic in this growing market sector".

Please check the literature references in the article so that their content is consistent with what the article is about.

The conclusion part requires changes.

Author Response

Thank you very much for taking the time to review this manuscript!

Response to Reviewer 2 Comments

1. Summary

Thank you very much for taking the time to review this manuscript. Please find the detailed responses below and the corresponding revisions in track changes in the re-submitted files.

2. Questions for General Evaluation

Reviewer’s Evaluation

Response and Revisions

Does the introduction provide sufficient background and include all relevant references?

Must be improved

 We appreciate your feedback on the introduction section of our manuscript. We have revised the introduction part, and we have carefully reviewed our sources to ensure we have included all relevant references. We believe these changes will improve the clarity and comprehensiveness of our introduction.

Are all the cited references relevant to the research?

Must be improved

Thank you for bringing our attention to the relevance of our cited references. We have thoroughly reviewed all the references and removed any that were not directly pertinent to our research. We have also added additional references that are more relevant and provide greater support to our study.

Is the research design appropriate?

Can be improved

We appreciate your feedback on our research design. We have reconsidered our approach and made necessary modifications to ensure the design is robust and appropriate for our study. We believe these adjustments will strengthen our research methodology and results.

Are the methods adequately described?

Can be improved

We value your comment on the description of our methods. We have revised this section to provide more detailed explanations of our methods, ensuring that our process is clear and replicable.

Are the results clearly presented?

Can be improved

Thank you for your suggestion to improve the presentation of our results. We have revised this section to ensure our results are clearly articulated and easily understood.

Are the conclusions supported by the results?

Must be improved

We appreciate your feedback on our conclusion. We have revised this section to ensure that our conclusions are strongly supported by and accurately reflect our results.

3. Point-by-point response to Comments and Suggestions for Authors

Comments 1: Lines 57-58. Incomprehensible statement: "an important yet under-re-searched demographic in this growing market sector".

Response 1: Thank you for pointing this out. We agree with this comment. Therefore,we have removed this sentence to make the paragraph more coherent.

Comments 2: Please check the literature references in the article so that their content is consistent with what the article is about.

Response 2: Thank you for your valuable feedback. We have carefully reviewed and revised the references, particularly in the first section of the literature review on plant-based milk market, to ensure the references directly correlate with the cited content. Additionally, we have revised the literature review in the second paragraph of the discussion part, removing some less relevant sources to maintain the consistency and focus of our study.

Comments 3: The conclusion part requires changes.

Response 3: Thank you for your comment. We made some necessary changes in conclusion part as follows:

This paper examined consumers' preferences and WTP for PBM based on DBDC-CVM method. The findings imply that attributes such as the appearance, taste, nutritional value, environmental benefits of PBM, alongside consumers' environmental awareness, health consciousness and food neophobia significantly influence their preferences and WTP for PBM. Consumers with high environmental awareness are more likely to perceive PBM as environmentally friendly and are willing to pay a higher price for it. Similarly, consumers with high health consciousness tend to value the environmental benefits of PBM and prioritize purchase convenience, as it aligns with their health-conscious lifestyle, leading to a higher WTP for PBM. Interestingly, consumers with high food neophobia exhibit greater WTP for PBM than those with low or medium food neophobia, primarily due to their elevated taste expectations and the importance they place on purchase convenience. These results underscore the potential role of PBM in constructing a healthy and sustainable food system.

We hope that these revisions will enhance the overall quality of our manuscript and make the conclusion more comprehensive and impactful. Your suggestions and comments are invaluable in this refinement process. Thank you once again for your constructive feedback. We look forward to any additional comments or suggestions that you may have.